# Synthesis, Structure–Property Evaluation and Biological Assessment of Supramolecular Assemblies of Bioactive Glass with Glycyrrhizic Acid and Its Monoammonium Salt

**DOI:** 10.3390/ma15124197

**Published:** 2022-06-13

**Authors:** Alimjon D. Matchanov, Rakhmat S. Esanov, Tobias Renkawitz, Azamjon B. Soliev, Elke Kunisch, Isabel Gonzalo de Juan, Fabian Westhauser, Dilshat U. Tulyaganov

**Affiliations:** 1Institute of bioorganic Chemistry, Mirzo Ulugbek, 83, Tashkent 100170, Uzbekistan; olim_0172@mail.ru (A.D.M.); esanovr@mail.ru (R.S.E.); 2Department of Chemistry, National University of Uzbekistan, University Street 4, Tashkent 100174, Uzbekistan; 3Department of Orthopaedics, Heidelberg University Hospital, Schlierbacher Landstraße 200a, 69118 Heidelberg, Germany; direktion.orthopaedie@med.uni-heidelberg.de (T.R.); elke.kunisch@med.uni-heidelberg.de (E.K.); fabian.westhauser@med.uni-heidelberg.de (F.W.); 4Department of Natural-Mathematical Sciences, Turin Polytechnic University in Tashkent, 17, Small Ring Street, Tashkent 100095, Uzbekistan; 1136001@gmail.com; 5Disperse Solid Materials, Technical University Darmstadt, Otto-Berndt-Straße 3, 64287 Darmstadt, Germany; isabel.gonzalo@tu-darmstadt.de

**Keywords:** bioactive glasses, glycyrrhizic acid, supramolecular assemblies, bone tissue engineering

## Abstract

Medical nutrients obtained from plants have been used in traditional medicine since ancient times, owning to the protective and therapeutic properties of plant extracts and products. Glycyrrhizic acid is one of those that, apart from its therapeutic effect, may contribute to stronger bones, inhibiting bone resorption and improving the bone structure and biomechanical strength. In the present study, we investigated the effect of a bioactive glass (BG) addition to the structure–property relationships of supramolecular assemblies formed by glycyrrhizic acid (GA) and its monoammonium salt (MSGA). FTIR spectra of supramolecular assemblies evidenced an interaction between BG components and hydroxyl groups of MSGA and GA. Moreover, it was revealed that BG components may interact and bond to the carboxyl groups of MSGA. In order to assess their biological effects, BG, MSGA, and their supramolecular assemblies were introduced to a culture of human bone-marrow-derived mesenchymal stromal cells (BMSCs). Both the BG and MSGA had positive influence on BMSC growth, viability, and osteogenic differentiation—these positive effects were most pronounced when BG1d-BG and MSGA were introduced together into cell culture in the form of MSGA:BG assemblies. In conclusion, MSGA:BG assemblies revealed a promising potential as a candidate material intended for application in bone defect reconstruction and bone tissue engineering approaches.

## 1. Introduction

Medical nutrients obtained from plants have been used in traditional medicine since ancient times, owning to the protective and therapeutic properties of plant extracts and products. Glycyrrhizic acid (GA) is one of those medical nutrients. GA is a triterpenoid saponin glycoside. The amphiphilic nature of the GA and its salts, such as monoammonium salt (MSGA), consisting of hydrophobic (triterpene) and hydrophilic (carbohydrate) parts (Figure 1), manifests itself in a number of its chemical properties, including its surface-active and gel-forming properties. Therefore, it can be used as a solubilizer for many water-insoluble drugs. For example, practically water-insoluble hydrocortisone, prednisolone, uracil, nystatin, and other drugs in combination with GA form aqueous solutions [1]. Moreover, glycyrrhizin is a particularly interesting candidate as a potential broad-spectrum antiviral agent [2,3,4,5].

Apart from its therapeutic effects, GA has shown to inhibit bone resorption and improve bone structure and biomechanical strength [6,7]. This feature of GA may potentially contribute to the healing of surgically created defects on bone tissue, where bioactive glass is used as bone graft material. For example, in cases of osteoporotic bone, a combination of bioactive bone substitute materials such as bioactive glasses (BG) and GA might be used to repair bone defects. Since the development of the 45S5-BG (composition in wt.%: 45 SiO_2_, 24.5 CaO, 24.5 Na_2_O, 6 P_2_O_5_) in the late 1960s by Hench and coworkers, the family of BGs grew rapidly, making BGs a major player, at least in experimental bone tissue engineering applications [8,9]. In the last few years, our groups have designed, developed, and evaluated the BG1d-BG (composition in wt.%: 46.1 SiO_2_, 28.7 CaO, 8.8 MgO, 6.2 P_2_O_5_, 5.7 CaF_2_, 4.5 Na_2_O). This BG was found to be biocompatible, not only upon implantation into rabbit femurs, but also when used for the treatment of jawbone defects in 45 human patients [10]. Furthermore, compared to the 45S5-BG composition, the BG1d-BG showed increased promotion of viability and proliferation in human bone-marrow-derived mesenchymal stromal cells (BMSCs) [11].

It is well documented that bioactive glass indices specify biological activity. They promote the release of soluble ionic species and bonds strongly with both soft and hard tissues due to their ability to form hydroxyapatite precipitation (HA) in contact with physiological fluids. The bioactivity mechanism includes a specific step at which bioactive glasses spontaneously expose hydroxyl (-OH) groups upon contact with water-based media. This can be exploited also for the effective grafting of specific moieties to their surface [12,13,14]. In this regard, an interesting and versatile strategy has recently emerged, namely the surface functionalization of bioactive glasses. It combines the well-known properties of bioactive glasses with the specific properties of the grafted moieties designed for selected applications. This is achieved through the addition of active biomolecules that can be grafted directly to the glass surface or by means of proper spacers. As it was widely reported, the surface grafting of active molecules does not inhibit the glass bioactivity [15,16,17]. For instance, peptides can be grafted to bioactive glasses using dopamine as a coupling agent, to get antifouling surfaces, while the peptide–surface interactions can be due to the van der Waals forces, such as the hydrogen bond, electrostatic attraction, hydrophobic effect, etc [18,19].

In this context, this work aims to synthesize and evaluate structure–property relationships of the supramolecular assemblies of bioactive glass with GA and MSGA. Since the aqueous solubility of glycyrrhizic acid is relatively low, the particular interest was addressed to complexes with MSGA that is more soluble in water [20]. The structural characterization of MSGA:BG was performed by means of Fourier transform infrared and UV–visible spectroscopies. Finally, MSGA:BG assemblies were introduced to a culture of BMSCs in order to assess cytocompatibility, as well as their influence on proliferation and osteogenic differentiation.

## 2. Materials and Methods

### 2.1. Synthesis of BG

BG with nominal composition (mol%) 4.33, Na_2_O—30.30, CaO—12.99, MgO—45.45, SiO_2_—2.60, P_2_O_5_—4.33, and CaF_2_ was synthesized from high-purity powders of SiO_2_ (purity > 99.5%), CaCO_3_ (>99.5%), MgCO_3_ (BDH Chemicals Ltd., London, UK, purity > 99.0%), Na_2_CO_3_ (Sigma Aldrich, Lisboa, Portugal, >99.0%), NH_4_H_2_PO_4_ (Sigma Aldrich, Darmstadt, Germany, >99.0%), and CaF_2_ (Sigma Aldrich, Germany, 325 mesh, >99.9%). A homogeneous mixture of the batch (100 g), obtained by ball milling, was preheated at 900 °C for 1 h for calcination, and then melted in Pt crucibles at 1400 °C for 1 h in air. Glass frits were produced by the quenching of the melts into cold water. The frits were dried and then milled for 2 h in a high-speed planetary mill (Pulverisette 6, Fritsch, Germany) and passed through a 32 µm sieve to obtain the final particles below 32 μm.

### 2.2. Preparation of GA and MSGA

MSGA was obtained from the licorice root of Glycyrrhiza glabra at the Institute of Bioorganic Chemistry, Academy of Sciences of Republic of Uzbekistan, through the following route: at the first stage, shredded roots of Glycyrrhiza glabra underwent extraction by 0.5–1.0 wt.% aqueous ammonium hydroxide solution at 90–105 °C, then precipitated by concentrated H_2_SO_4_, followed by re-extraction by acetone. The next stages of the route included: (a) the precipitation of triammonium salt of GA by 25 wt.% ammonia solution, and (b) its transformation into the monoammonium salt of GA through the crystallization from glacial CH_3_COOH. Afterwards, this product was recrystallized from ethanol to get MSGA with a purity of 85–88%. Finally, treatment of MSGA by 1 wt.% H_2_SO_4_ yielded GA with a purity of 93–95%.

### 2.3. Synthesis GA:BG and MSGA:BG. Supramolecular Assemblies of Bioactive Glass with Glycyrrhizic Acid and Its Salt

Bioactive glass powder was introduced to an aqueous–alcoholic solution (water–alcohol ratio 50/50 by volume) of GA or MSGA at different weight ratios, as 1:10, 1:20, and 1:50, under vigorous stirring for 5–6 h at room temperature. The organic part was distilled off on a rotary evaporator and the aqueous part was freeze-dried. The final product was obtained in the form of a powder featuring a creamy color with possible gray splashes (Figure 2). Overall, 6 compositions were synthesized, namely GA:BG 50:1; GA:BG 20:1; GA:BG 10:1 and MSGA:BG 50:1; MSGA:BG 20:1; MSGA:BG 10:1. Gels were prepared from the experimental powders by adding water in the ratio of 1 wt.% of the powder to 99 wt.% of water, followed by vigorous stirring.

### 2.4. Structural Characterization of the Supramolecular Assemblies

Infrared spectra of the supramolecular assemblies in the form of powders were obtained using a Fourier transform infrared spectrometer (PerkinElmer Spectrum IR). For this purpose, samples were crushed to powder form, mixed with KBr in the proportion of 1/150 (by weight), and pressed into a pellet using a hand press.

UV absorption spectra of the supramolecular assemblies were recorded using Shimadzu-1280 (Japan), pH of 0.1 wt.%, and solutions were measured using pH meter KSL-1100-1.

### 2.5. Investigation of the Biocompatibility of MSGA:BG Supramolecular Assemblies of BG and MSGA Using BMSCs

#### 2.5.1. Isolation and Cultivation of BMSCs

Human BMSCs were isolated from bone marrow, as described previously [21,22]. In short, BMSCs were obtained by a Ficoll-Paque Plus density gradient (GE Healthcare Europe, Freiburg, Germany). Cells were cultivated in 0.1% gelatin-coated cell culture flasks (Nunc, Roskilde, Denmark) with expansion medium composed of Dulbecco’s Modified Eagle’s Medium (DMEM), high glucose, supplemented with 12.5% Fetal Calf Serum (FCS), 1% L-glutamine, 1% nonessential amino acids (NEAA; all Life Technologies, Dreieich, Germany), 1% penicillin/streptomycin (Biochrom, Berlin, Germany), 0.1% β-mercaptoethanol (Life Technologies), and 4 ng/mL fibroblast growth factor 2 (Active Bioscience, Hamburg, Germany). Medium was exchanged after the first 24 h to discard nonadherent cells, and subsequently three times a week. Reaching 80% confluency, the cells were frozen in liquid nitrogen until further use. For cell culture experiments, isolated BMSCs from 10 patients were pooled to minimize interpatient variability [21]. Written consent was obtained from the patients prior to collection of the bone marrow. The study was approved by the responsible ethics committee of the Medical Faculty of Heidelberg University (S-340/2018).

To analyze the influence of supramolecular assemblies of BG and MSGA on cell number, cell viability, and alkaline phosphatase (ALP) activity of BMSCs, cells were seeded in the presence of MSGA, BG, or the supramolecular assemblies of both in microtiter plates in a density of 1.9 × 10^4^ cells/cm^2^. For the experiments, stock solutions of MSGA, MSGA:BG supramolecular assemblies, and BG were prepared under sterile conditions, and further diluted to a concentration of 1 mg/mL for MSGA and the supramolecular assemblies of MSGA:BG, and 0.1 mg/mL for BG. In all cell culture experiments, DMEM, 10% FCS, and 1% penicillin/streptomycin was used. Cell culture medium, MSGA, and BG-treated groups served as controls.

#### 2.5.2. Determination of Cell Number by DAPI Quantification

A 4′,6-diamidino-2-phenylindole (DAPI) staining protocol [23] was used to indirectly analyze the impact of MSGA:BG supramolecular assemblies on BMSCs proliferation by DNA quantification. In brief, after the indicated time points, cells were washed once with DPBS and fixed with 70% ethanol for 10 min. Nuclei of the cells were stained with DAPI (Invitrogen; 1 µg/mL in buffer solution) for 5 min. Excessive dye was removed with a wash solution (0.5 M NaCl, 0,2% Tween-20, and 20 mM citrate buffer, pH 5). For quantification, nuclear-bound DAPI was eluted using elution buffer (2% sodium dodecyl sulphate, 20 mM Tris-HCl, pH 7) for 15 min at RT. Eluted dye was transferred to a white 96-well microtiter plate and DAPI fluorescence was measured using a Wallac 1420 Victor 2 microplate reader (Perkin Elmer, Waltham, MA, USA).

#### 2.5.3. Analysis of Cell Viability and Activity of Alkaline Phosphatase

Cell viability and ALP activity was analyzed, as previously described [24]. In brief, cell culture supernatants were removed, and cells were washed once with DPBS, followed by an incubation step with fluorescein diacetate (FDA, 0.002 mg/mL in DPBS; Carl Roth, Karlsruhe, Germany) for 5 min at 37 °C. After a washing step with DPBS, cells were lysed using 0.5% Triton-X-100 (Sigma-Aldrich) for 5 min at 37 °C. Aliquots of the cell lysates were transferred to a white 96-well microtiter plate, and the ALP substrate 4-methylumbelliferone phosphate (MUP; Thermo Fisher, Waltham, MA, USA) was added and incubated for 15 min at 37 °C. Fluorescence intensities of FDA and 4-methylumbelliferone were measured using a Wallac 1420 Victor 2 microplate reader (Perkin Elmer). A dilution series of shrimp ALP (Thermo Fisher) ranging from 50 mU to 0.15 mU was run in parallel to calculate the ALP activity. ALP activity was normalized to DAPI fluorescence intensity. Cell viability was calculated by normalization of the FDA fluorescence intensities to the mean of the DAPI fluorescence intensities. Thereafter, the DMEM control of each time point was set as 100%. A viability of 70% was set as threshold.

### 2.6. Statistics

Statistics were calculated using the IBM SPSS (Version 25; IBM, Armonk, NY, USA). Significances were tested with the nonparametric Kruskal–Wallis test, followed by the Mann–Whitney U test. Differences were considered statistically significant for *p* ≤ 0.05. All results were expressed as means ± SD.

## 3. Results

### 3.1. Structural Features of Supramolecular Assemblies MSGA:BG and GA:BG

A pH of the as-prepared 0.1 wt.% aqueous solutions of supramolecular assemblies are presented in Table 1.

Figure 3 shows the FTIR spectra of the supramolecular assemblies, along with the MSGA and GA spectra presented as controls. Thus, the FTIR spectrum of the MSGA and GA exhibited transmittance bands attributed to valent vibrations of the (-OH) groups in the interval of wavenumbers of 3399–3249 cm^−1^ and 3423–3207 cm^−1^, respectively. Valent vibrations of carbonyl groups (C = O) were revealed at 1715 cm^−1^ (MSGA) and at 1722 cm^−1^ (GA). Deformation vibrations of methyl and methylene groups were revealed at 1455, 1365, 1262, and 1215 cm^−1^ and at 1454, 1361, and 1305, and 1259, 1211, and 1166 cm^−1^ for MSGA and GA, respectively.

A comparison of the FTIR spectrum of MSGA with the spectra of the MSGA:BG complexes revealed changes in the intensity and position of bands in the region of 3500–3150 cm^−1^ (Figure 3a). In particular, BG incorporation caused a lack of sharp features of OH-stretching vibrations, which is evidence of the interaction between the BG components (e.g., cations of Na^+^, Ca^2+^, and Mg^2+^) and hydroxyl groups of saponin. Conversely, the bending vibrations of the methyl groups at 2950–2850 cm^−1^ do not undergo significant changes. However, the changes are apparent in the range of vibrations of the carbonyl functional groups of carboxyl: the vibration signals of MSGA at 1750–1640 cm^−1^ significantly decreased in intensity, while their peak maxima shifted to the lower wavenumbers by 40–55 cm^−1^, indicating that the BG components may interact and bond to the carboxyl of saponin as well (Figure 3a).

Similar features were observed when the FTIR spectra of GA were compared with the spectra of the GA: BG complexes (Figure 3b). The main differences were revealed in the region of the stretching vibrations of the hydroxyl groups at 3600–3200 cm^−1^, where the spectra of the GA: BG complexes demonstrated a broad shoulder. Bending vibrations of the methyl groups at 2950–2850 cm^−1^ remained unaffected, and, in contrast to MSGA:BG, no significant changes were observed in the range of vibrations of the carbonyl groups in the interval of wavenumbers of 1750–1640 cm^−1^.

UV spectra of the supramolecular complexes of MSGA:BG (Figure 4a) and GA:BG (Figure 4b) were obtained at concentrations of 0.1 mg/mL in 50% ethanol. A characteristic peak of saponin absorption is observed in the near UV region of the spectrum at 252–255 nm, corresponding to the π → π * transition of the conjugated bond in the C ring and C = O group. Since this part of the molecule (i.e., the region of conjugated bond in the C ring and C = O group at the triterpene part) remains unabated upon complexes formation, there is no change in the peak absorption in the UV spectra of saponin.

### 3.2. Influence of MSGA:BG Complexes on Cell Number, Cell Viability, and ALP Activity

The influence of the MSGA:BG complexes on the cell number, cell viability, and ALP activity was analyzed using BMSCs. The cell number in the group treated with the 10:1 MSGA:BG complex was equal or even higher compared to one of the control groups (control, MSGA, or BG group), indicating a positive influence on the cell number of this mixture (Figure 5a). In contrast, the cell number in the 50:1 MSGA:BG-treated group was significantly lower compared to the control groups (control, MSGA, or BG group) on nearly all analyzed time points, indicating a negative impact. When comparing the MSGA:BG-treated groups, the group treated with the 10:1 MSGA:BG complex had a significantly higher cell number compared to the group treated with the 50:1 complex on all analyzed time points, indicating a ratio dependent influence of the MSGA:BG complexes on the cell number.

Analyzing the influence of the MSGA, BG, and MSGA:BG complexes on cell viability showed that nearly all treated groups reached a cell viability of 70% or even higher, except for the 50:1 complex-treated group on day 7 (viability: 60.3% ± 10.5; Figure 5b). As observed for the cell number, the viability of the cells treated with the 10:1 complex was equal or even higher compared to at least one of the control groups (control, MSGA, or BG group) on all analyzed time points, except for day 14. Moreover, the viability of the 50:1 complex-treated cells were significantly lower compared to at least one of the control groups (control, MSGA, or BG group). Comparing the viability between the MSGA:BG-treated groups, again the group treated with the 10:1 MSGA:BG complex had significantly better viability compared to the group treated with the 20:1 and 50:1 complex on all analyzed time points, also indicating a ratio-dependent influence of the MSGA:BG complexes on the cell viability.

Notably, a significantly higher ALP activity was observed in BMSCs treated with the 10:1 complex, as well as the cells treated with the 50:1 complex, compared to at least one of the control groups (control, MSGA, or BG group; Figure 5c). Additionally, between the MSGA:BG-treated groups, the 10:1 complex-treated cells and the 50:1-treated cells had a significantly higher ALP activity compared to the 20:1-treated cells.

## 4. Discussion

Nowadays, plant-based antimicrobial compounds are a safer alternative to the chemicals used in the treatment of dental plaque and oral diseases [25,26,27,28,29]. In particular, glycyrrhizic acid (GA) is one of the most bioactive ingredients of licorice plants that stimulate saliva due to its sweet taste and protect from dental caries [30]. An amphiphilic molecule of GA can form host–guest complexes or micelle-type nanocarriers by self-assembly, thereby encapsulating the guest molecule and increasing its solubility [20,31]. The use of various NMR techniques allows to demonstrate the aggregation processes of GA in a water/methanol mixture [32]. The interaction between MSGA and bovine serum albumin was investigated by fluorescence and absorption spectroscopy [33]. The structure of supramolecular assemblies formed by GA/MSGA with salicylic acids and diterpenoid lagochiline was studied using FTIR/UV and NMR [34,35].

In the present work, a spectroscopic assessment of the interactions between BG components and MSGA/GA was performed by means of FTIR and UV techniques. The bands for the unreacted glass powder are mostly owing to the Si–O vibrational modes at the low-frequency region, with wavenumbers in the interval of 1100–400 cm^−1^. In particular, the bands around 1040 cm^−1^ and 930 cm^−1^ indicate the distribution Q^3^ and Q^2^ silicate units, respectively, while the band near 1030 cm^−1^ is attributed to the PO_3_ end-groups due to the presence of P_2_O_5_ [36]. Importantly, some silica–oxygen and phosphorus–oxygen vibrations in BG could be masked by the stretching vibrations of the saponin’s C–O–C and C–OH groups in the region of 1200–1000 cm^−1^. The most striking change, however, was observed when compared to the spectra of the pure MSGA and GA with the FTIR spectra of the MSGA:BG and GA:BG complexes at 3600–3200 cm^−1^ that belong to the region of stretching vibrations of hydroxyl groups (Figure 3). In particular, BG incorporation caused a lack of sharp features of OH stretching vibrations and the appearance of a broad shoulders (Figure 3a,b). Moreover, the MSGA:BG assemblies’ vibration signals of MSGA at 1750–1640 cm^−1^ significantly decreased in intensity, and the peaks maxima shift to the lower wavenumbers by 40–55 cm^−1^ These features indicate that BG structural units interact and bond to the hydroxyl and carboxyl groups of saponin (Figure 3). In contrast, according to the UV spectroscopy data (Figure 4), the part of the molecule at the hydrophobic (triterpene part) remains unabated.

From Table 1, the acidity resulting from the carboxylic groups of MSGA and GA increases with addition of the BG components. The rise in the pH when bioactive glasses are immersed in aqueous solution is a well-known phenomenon. This can be explained by water molecules penetrating the silicate network and directly influencing the ion exchange between the sodium and calcium cations of the BG and hydrogen cations from the dissolution medium. As a result, modifier ions are partially substituted for hydrogen cations, which bind to nonbridging oxygens (NBO) and form silanol groups (Si–OH), while the dissolution medium becomes depleted of protons, causing the pH to increase [36,37]. Thus, the results of the dissolution experiments of BG in the SBF solution are discussed elsewhere [10].

Linkage within MSGA/BG and GA/BG assembles can be achieved through the intermolecular forces as hydrogen bonding by using the hydroxyl (and/or carboxyl) functional groups of saponin and structural units of BG at its surface. A proposed scheme of interaction between MSGA and BG in the as-prepared composites is presented in Figure 6. The above-mentioned features allow us to classify MSGA/BG and GA/BG materials as Class I hybrids that feature weak bonds, such as electrostatic, van der Waals, and hydrogen bonds between an inorganic and an organic part [38,39,40].

The properties of the hybrid differ from the individual properties of the two materials because of the interactions between them. Therefore, the MSGA:BG assemblies, along with BG, were introduced to a culture of BMSCs in order to assess the cytocompatibility, as well as their influence on proliferation and osteogenic differentiation. In a recently conducted study by our groups, the BG1d-BG composition already demonstrated favorable biocompatibility and enhanced the osteogenic differentiation of BMSCs [11]. In this study, BG1d alone proved its ability to enhance BMSC growth, viability, and osteogenic differentiation compared to BMSCs that were cultivated without the presence of either BG or MSGA and its derivates. The introduction of MSGA to the cell culture did not affect their viability or growth when compared to the control group—similar findings were recently described by Bai and coworkers [41]. Similar to the findings reported in their study, MSGA alone also enhanced ALP activity in our study [41] These osteostimulative properties could even be further improved in the MSGA:BG assemblies. An enhancement of osteogenic properties of bone-substitute materials by BGs has been described before, for example, for combinations of calcium phosphates and 45S5-BG [42,43]. In order to understand the role of MSGA and BG1d in greater detail, further analyses must be conducted, such as fluorescence microscopy, to visualize cell growth patterns within the gels, gene expression analyses focusing, e.g., on osteogenic differentiation, pathway analyses, e.g., via Western blotting and/or the analysis of the properties of the MSGA:BG under in vivo conditions, for example, in bone defect models [44].

## 5. Conclusions

Supramolecular assemblies of bioactive glass with GA and MSGA were synthesized. The final products were obtained in the form of fine powders or gels. The pH change in 0.1 wt.% aqueous solution is correlated with the content of bioactive glasses that introduces modifier ions which are partially substituted for hydrogen cations, resulting in a reduced number of protons in the dissolution medium. Spectroscopic studies reveled that the interactions between BG components and MSGA/GA occur at the hydrophilic (carbohydrate part) of the saponin molecule. Notably, linkage within MSGA/BG and GA/BG assembles can be achieved through the intermolecular forces as hydrogen bonding by using the hydroxyl (and/or carboxyl) functional groups of saponin and the structural units of BG at its surface. Both BG1d-BG and MSGA had positive influence on BMSC growth, viability, and osteogenic differentiation—these positive effects were most pronounced when BG1d-BG and MSGA were introduced together into the cell culture in the form of MSGA:BG assemblies. In conclusion, the MSGA:BG assemblies revealed a promising potential as candidate materials intended for application in bone defect reconstruction and bone tissue engineering approaches.

## Figures and Tables

**Figure 1 materials-15-04197-f001:**
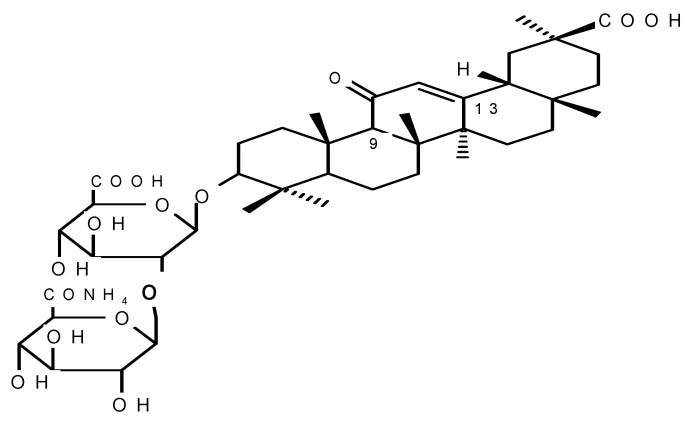
Structure of monoammonium salt of glycyrrhizic acid (MSGA): the right upper part is hydrophobic (triterpene part), and the left is hydrophilic (carbohydrate part).

**Figure 2 materials-15-04197-f002:**
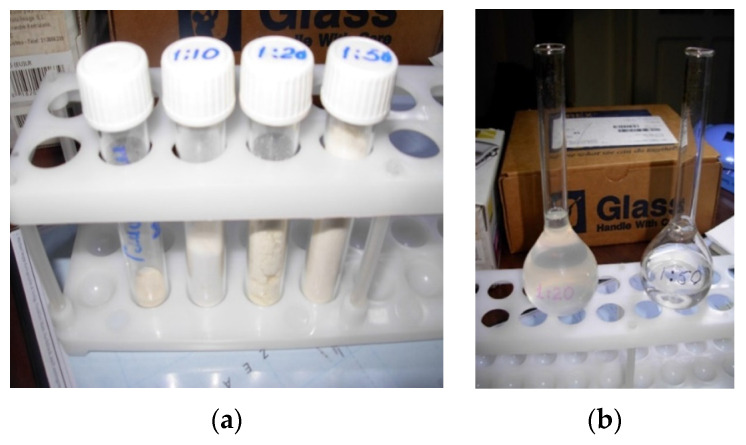
Photographs of supramolecular assemblies of bioactive glass with MSGA: (**a**) in powder form, from the right to the left samples MSGA:BG 50:1; MSGA:BG 20:1; MSGA:BG 10:1 and the leftmost is MSGA; (**b**) in gel form, from the left to the right samples MSGA:BG 20:1 and MSGA:BG 50:1.

**Figure 3 materials-15-04197-f003:**
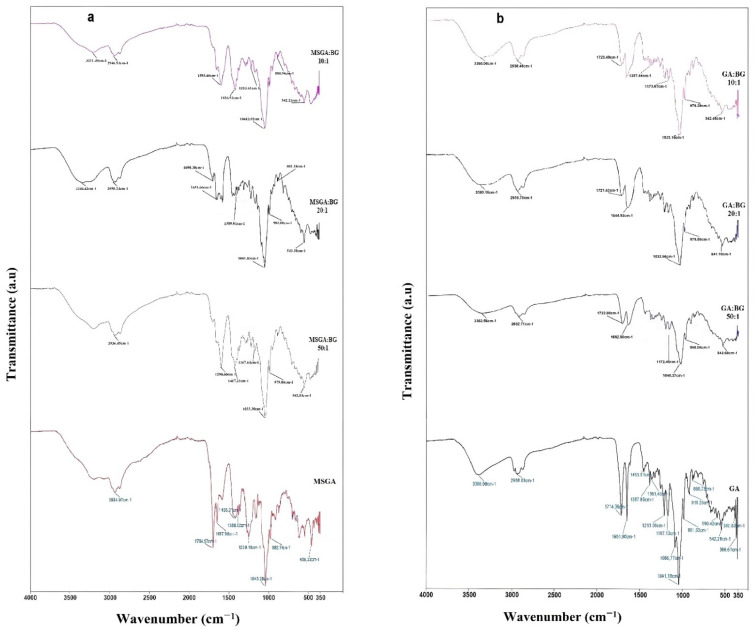
FTIR spectra of (**a**) MSGA and MSGA:BG complexes, (**b**) GA and GA:BG complexes.

**Figure 4 materials-15-04197-f004:**
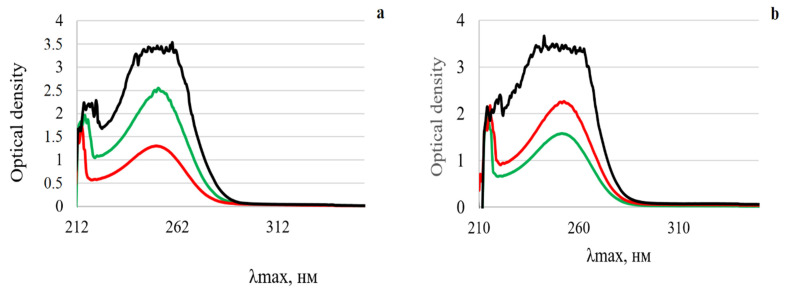
UV spectra of (**a**) MSGA:BG complexes and (**b**) GA:BG complexes: lines in red—MSGA/GA:BG 10:1, lines in green—MSGA/GA:BG 20:1, lines in black—MSGA/GA:BG 50:1.

**Figure 5 materials-15-04197-f005:**
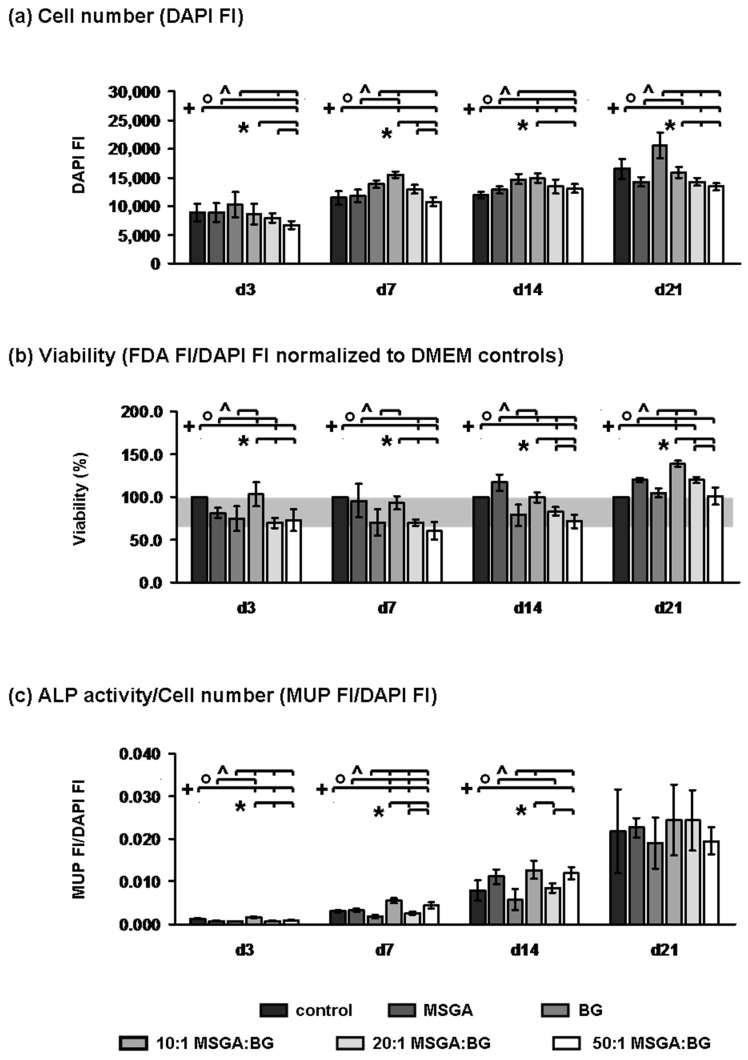
Impact of supramolecular complexes of MSGA:BG on cell number, viability, and ALP activity of BMSCs. Bars show mean ± SD. *p* < 0.05 Mann–Whitney U test control (+), MSGA (°), or BG (^) group vs. MSGA:BG complex-treated groups, as indicated by brackets; * *p* < 0.5 Mann–Whitney U test between the MSGA:BG complexes treated groups as indicated by brackets. The grey box in (**b**) shows the range of 70% (threshold) to 100% viability. FI: fluorescence intensity.

**Figure 6 materials-15-04197-f006:**
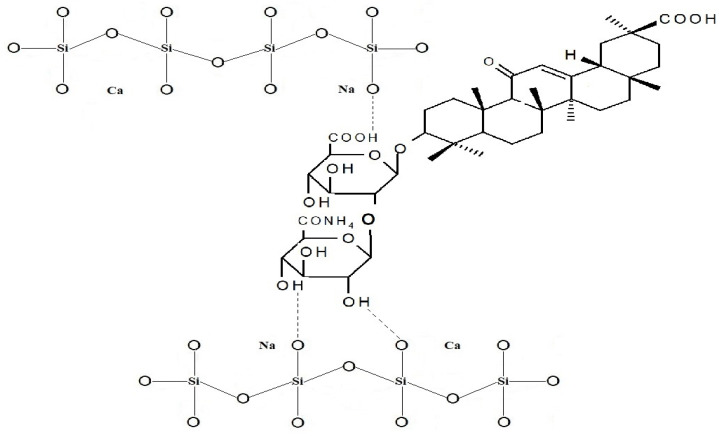
Proposed scheme of interaction between MSGA and BG’s structural units in the as-prepared hybrids.

**Table 1 materials-15-04197-t001:** pH of as-prepared 0.1 wt.% solutions.

No.	Supramolecular Assemblies	pH
1	GA:BG 10:1	3.29
2	GA:BG 20:1	3.23
3	GA:BG 50:1	2.99
4	MSGA:BG 10:1	4.33
5	MSGA:BG 20:1	3.92
6	MSGA:BG 50:1	3.43

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
