# Peer review of "Synthesis, Structure–Property Evaluation and Biological Assessment of Supramolecular Assemblies of Bioactive Glass with Glycyrrhizic Acid and Its Monoammonium Salt"

_materials, 2022, doi:10.3390/ma15124197_

Round 1
Reviewer 1 Report
I would like to thank the authors for their Article entitled "Synthesis, structure-property evaluation and biological assess- 2 ment of supramolecular assemblies of bioactive glass with 3 glycyrrhizic acid and its mono-ammonium salt." It is well written, informative and intersting to the readers.
I only have limited comments in line 56 in introduction section the phrase bioactive glasses (BG) and GA might be used to regenerate bone defects. can be changed to repair bone defects or regenerate bone tissue.
Line 190 the results is conjugated with discussion although there is a separate discussion section.
References should be revised for a standardized style
Author Response
Response to the Reviewers’ comments
Reviewer 1.
Comments and Suggestions for Authors
I would like to thank the authors for their Article entitled "Synthesis, structure-property evaluation and biological assessment of supramolecular assemblies of bioactive glass with 3 glycyrrhizic acid and its mono-ammonium salt." It is well written, informative and intersting to the readers.
I only have limited comments in line 56 in introduction section the phrase bioactive glasses (BG) and GA might be used to regenerate bone defects. can be changed to repair bone defects or regenerate bone tissue.
Answer: Thank you for this comment. We corrected the corresponding sentence.
Line 190 the results is conjugated with discussion although there is a separate discussion section.
Answer: Thank you for this comment. We made corresponding correction.
References should be revised for a standardized style
Answer: Thank you for this comment. The references have been revised for a standardized style.
Reviewer 2 Report
materials-1750385
Title: Synthesis, structure-property evaluation and biological assessment of supramolecular assemblies of bioactive glass with glycyrrhizic acid and its mono-ammonium salt
In this manuscript, glycyrrhetinic acid and its monoammonium salt were immobilised on the surface of novel bioactive glass particles developed by the authors' research. results on the growth, viability and alkaline phosphatase activity of BMSCs indicate the potential application of MSGA/GA:BG in bone defect reconstruction and bone tissue engineering.
Here are my comments:
1. English expression need to be improved.
2. Page 5, line 203-213. The hydroxyl and carbonyl peaks were not attenuated by the interaction with BG, but more likely by the huge difference in amount between surface-immobilised MSGA and pure MSGA resulting in a weakening of the peak intensity. In fact, the intensity of all the peaks associated with MSGA and GA in the spectra of supramolecular assemblies were diminished.
3. Page 6, line 214-220. Please analyse in more detail the absorption peaks of the silica-oxygen and phosphorus-oxygen structures in BG.
4. Page 6, line 221-222. The figure is not clear to see, and the x-axis is not consistent.
-
5. Page 6-7, line 230-233. Neat MSGA and GA spectra should be presented here as controls.
6. Page 8, line 258-262. Why were MSGS:BG and GA:BG mixed together in culture and cytologically characterized?
7. Page 14, line 332-335. Have you measured the pH of neat MSGA/GA aqueous solutions? The pH of MSGA/GA:BG aqueous solutions decreased because of a decrease in BG content, but it could also be due to an increase in MSGA/GA content as they are acidic.

Author Response
Reviewer 2
Comments and Suggestions for Authors
Title: Synthesis, structure-property evaluation and biological assessment of supramolecular assemblies of bioactive glass with glycyrrhizic acid and its mono-ammonium salt
In this manuscript, glycyrrhetinic acid and its monoammonium salt were immobilised on the surface of novel bioactive glass particles developed by the authors' research. results on the growth, viability and alkaline phosphatase activity of BMSCs indicate the potential application of MSGA/GA:BG in bone defect reconstruction and bone tissue engineering.
Here are my comments:
- English expression need to be improved.
Answer: We checked the expressions and corrected/adapted.
- Page 5, line 203-213. The hydroxyl and carbonyl peaks were not attenuated by the interaction with BG, but more likely by the huge difference in amount between surface-immobilised MSGA and pure MSGA resulting in a weakening of the peak intensity. In fact, the intensity of all the peaks associated with MSGA and GA in the spectra of supramolecular assemblies were diminished.
Answer: Thank you for this important comment that will be considered in our further studies.
- Page 6, line 214-220. Please analyse in more detail the absorption peaks of the silica-oxygen and phosphorus-oxygen structures in BG.
Answer: We added to the Discussion part more details related to the absorption peaks of the silica-oxygen and phosphorus-oxygen structures in BG.
- Page 6, line 221-222. The figure is not clear to see, and the x-axis is not consistent.
Answer: We apologies for this inconvenience. The new figure (i.e. Figure 3,a and 3,b) was prepared with consistent x-axes.
- Page 6-7, line 230-233. Neat MSGA and GA spectra should be presented here as controls.
Answer: Neat MSGA and GA spectra are presented as controls in the new figure (i.e. Figure 3,a and 3,b).
- Page 8, line 258-262. Why were MSGS:BG and GA:BG mixed together in culture and cytologically characterized?
Answer: In cell culture, only mixtures of MSGA and BG were tested. We apologize for the misleading labeling in Figure 5. This has been changed now in the new figure (Lines 250ff).
- Page 14, line 332-335. Have you measured the pH of neat MSGA/GA aqueous solutions? The pH of MSGA/GA:BG aqueous solutions decreased because of a decrease in BG content, but it could also be due to an increase in MSGA/GA content as they are acidic.
Answer: Thank you for this comment. pH of neat MSGA/GA aqueous solutions were not measured. However, we do agree with the Reviewer that pH of MSGA/GA:BG aqueous solutions could decrease due to an increase in MSGA/GA content as they are acidic.
Reviewer 3 Report
Dear authors,
I would like to thank you for submitting an original research article on entwined effects of bioactive glass and glycyrrhizic acid. Enclosed below are my comments on the present manuscript:
Abstract
1. Double check the grammar within the text (e.g., line 20 – verb “is” is missing and “contributed” after “may” should be in the infinite form (contribute))
Introduction; Materials and methods
1. Line 98: Please add for how long were the frits milled in a high-speed planetary mill?
2. Line 110: How was the purity of GA determined? HPLC?
3. Line 130: Please specify the FTIR method used (number of scans, resolution).
4. Line 160: Please specify which DMEM was used? High glucose or low glucose?
5. Section 2.5.1.: Please describe how the powders were sterilized and were they pretreated before the submersion with cells? Were the powders in direct contact with cells?
6. Please revise the grammar throughout the text (e.g., Line 39 (such as, for instance – one is not necessary); line 41 (“manifests itself in a number of its chemical properties” – needs reformulation, be careful with the length of the sentences”); line 58, 68 and 72 (commas are needed in the sentence structure) etc.)
Results and discussion
1. Please improve the resolution quality of the Figure 3 (a, b and c). As presented it is not possible to differentiate almost anything. Furthermore, I believe there is a typo in the name of the X axis, however I could be wrong as I do not see clearly even that.
2. Figure 4, I suggest to move the legend of the series from the title to the Figure itself, so it is easier to follow.
3. Section 3.1.: I would suggest to add the discussion on the changes that transpired between the MSGA:BG complexes and GA:BG complexes, not only in comparison to the pure GA or MSGA. Especially because later in the manuscript the compositions among themselves were compared and their effect.
4. Line 237, 239: Please add in the brackets what was the control group.
5. Line 245 – 251: Please add the reference value which is considered to be safe for the cells (i.e. ISO standard claiming that cell viability of 70% was set as a threshold) and connect it to the chosen units for representing the results (FDA fluorescence intensity) or express the results in % viability.
Discussion
1. Line 264 – 277: I would suggest to incorporate this part within the introduction. The discussion part should refer closely to the research done by the authors themselves not the general characteristics of the components in question.
2. Line 292 – 294: Please revise the grammar (use of commas and e.g., The pH rises when -> The rise in the pH, when…)
Conclusion
1. I would suggest to add the importance of testing osteo-specific gene and protein expression by real time quantitative PCR, western blotting and immunofluorescence analyses in the further research. ALP tested in the present manuscript is only one of the indicatives of cell differentiation.
General comment
Please revise the use of English language; especially pay attention to the use of commas.
Please mark all the changes in the manuscript with a color marker so it is easier to track them (including language revision).

Author Response
Reviewer 3
Comments and Suggestions for Authors
Dear authors,
I would like to thank you for submitting an original research article on entwined effects of bioactive glass and glycyrrhizic acid. Enclosed below are my comments on the present manuscript:
Abstract
- Double check the grammar within the text (e.g., line 20 – verb “is” is missing and “contributed” after “may” should be in the infinite form (contribute))
Answer: We checked the grammar and corrected the mistakes.
Introduction; Materials and methods
- Line 98: Please add for how long were the frits milled in a high-speed planetary mill?
Answer: The frits were milled for 2 hours in a high-speed planetary mill. Corresponding change was made in the section “Materials and Methods”.
- Line 110: How was the purity of GA determined? HPLC?
Answer: The purity of GA determined by HPLC was in the range of 93-95 %.
- Line 130: Please specify the FTIR method used (number of scans, resolution).
Answer: Infrared spectra were obtained using PerkinElmer Spectrum IR spectrometer with the resolution of 0.5см-1. At least 2 scans were performed for each sample. `
- Line 160: Please specify which DMEM was used? High glucose or low glucose?
Answer: DMEM with high glucose was used for the cell culture experiments. We added the missing information (page 6, line 142).
- Section 2.5.1.: Please describe how the powders were sterilized and were they pretreated before the submersion with cells? Were the powders in direct contact with cells?
Answer: Thank you for the comment. Indeed, we did not specify the handling of the powder and BG in the method sections. In pre-experiments and also in the experiments for the manuscripts, no contamination of the cultures was observed, although we did not sterilize the BGs and MSGA powders. They were just handled under sterile conditions and the media contained antibiotics (penicillin/streptomycin 1%). The powders and BG were introduced in the cell culture by preparing a stock solution of 10 mg/ml for MSGA and assemblies and 1 mg/ml of the BG powder. These stock solutions were further diluted to the used concentrations. We added the missing information to the method sections (page 6, line 152-155).
- Please revise the grammar throughout the text (e.g., Line 39 (such as, for instance – one is not necessary); line 41 (“manifests itself in a number of its chemical properties” – needs reformulation, be careful with the length of the sentences”); line 58, 68 and 72 (commas are needed in the sentence structure) etc.)
Answer: We checked the grammar and corrected the mistakes.
Results and discussion
- Please improve the resolution quality of the Figure 3 (a, b and c). As presented it is not possible to differentiate almost anything. Furthermore, I believe there is a typo in the name of the X axis, however I could be wrong as I do not see clearly even that.
Answer: We apologies for this inconvenience. The new figure (i.e. Figure 3,a and 3,b) was prepared with improved resolution and consistent x-axes.
- Figure 4, I suggest to move the legend of the series from the title to the Figure itself, so it is easier to follow.
Answer: We corrected Figure 4.
- Section 3.1.: I would suggest to add the discussion on the changes that transpired between the MSGA:BG complexes and GA:BG complexes, not only in comparison to the pure GA or MSGA. Especially because later in the manuscript the compositions among themselves were compared and their effect.
Answer: We thank the Reviewer for this valuable comment and would like to go deeper inside to the changes that transpired between the MSGA:BG and GA:BG complexes in our forthcoming study.
- Line 237, 239: Please add in the brackets what was the control group.
Answer: For reasons of clarity, we have summarized in the results part (chapter 3.2, page 11) the control, MSGA, and BG group as controls. We agree that this might be confusion, so we have now specified the different groups by adding brackets. In addition, for a better identification of these groups in figure 5, we have now assigned individual symbols for the control (+), MSGA (°), and BG (^) group. This is included in the new figure 5 and the figure legends (page 15, line 251).
- Line 245 – 251: Please add the reference value which is considered to be safe for the cells (i.e. ISO standard claiming that cell viability of 70% was set as a threshold) and connect it to the chosen units for representing the results (FDA fluorescence intensity) or express the results in % viability.
Answer: We thank the Reviewer for this valuable comment. We have recalculated the FDA data to present them as 100% viability of the DMEM controls (Fig. 5). The FDA fluorescence intensities were first normalized to the FDA fluorescence intensities (cell number). Thereafter, the values measured in the DMEM controls were set as reference (=100%). This recalculation of the cell viability showed that nearly all treated groups reached a cell viability of 70% or even higher except for the 50:1 complex-treated group on day 7 (viability: 60.3% ±10.5). We have now included this in the method section (Methods, page 7, line 178-179) and in the results part (Results, page 11, line 236-237). We exchanged the figure 5b to the new version showing recalculated viability data. We hope this presentation of the viability data fits the reviewers demands.
Discussion
- Line 264 – 277: I would suggest to incorporate this part within the introduction. The discussion part should refer closely to the research done by the authors themselves not the general characteristics of the components in question.
Answer: Thank you for this comment. We agree with the reviewer that the discussion part should refer closely to the research done by the authors themselves not the general characteristics of the components in question.
In fact, we thought that adding the above-mentioned part to the Discussion would allow readers compare and envisage the research techniques used to study supramolecular assemblies formed by GA/MSGA.
Finally, we omitted the second sentence from the Discussion part that bears the general characteristics of the components in question.
- Line 292 – 294: Please revise the grammar (use of commas and e.g., The pH rises when -> The rise in the pH, when…)
Answer: Thank you for this comment. We revised the grammar.
Conclusion
- I would suggest to add the importance of testing osteo-specific gene and protein expression by real time quantitative PCR, western blotting and immunofluorescence analyses in the further research. ALP tested in the present manuscript is only one of the indicatives of cell differentiation.
Answer: We agree with the Reviewer that analyzing the influence of the MSGA/BG assemblies on osteogenic differentiation of MSCs would be of great interest. However, including a huge data set about the influence of MSGA/BG assemblies on osteogenic differentiation would be going beyond the scope of the present manuscript which has the main focus on the synthesis and characterization of the MSGA/BG assemblies. Though, since this missing data is a potential limitation of the present study, we amended the discussion (Lines 309 f.).
General comment
Please revise the use of English language; especially pay attention to the use of commas.
Answer: We checked the expressions and corrected/adapted.
Please mark all the changes in the manuscript with a color marker so it is easier to track them (including language revision).
Answer: Thank you. We proceed according to your suggestion.
Round 2
Reviewer 3 Report
Dear Authors,
thank you for addressing the suggestions presented in the first review.